# Prevalence of hypertension and its determinants in Ethiopia: A systematic review and meta-analysis

**Sofonyas Abebaw Tiruneh**[1]*, **Yeaynmarnesh Asmare Bukayaw**[2], **Seblewongel Tigabu Yigizaw**[2], **Dessie Abebaw Angaw**[2]

**1** Department of Public Health, College of Health Sciences, Debre Tabor University, Debre Tabor, Ethiopia, **2** Department of Epidemiology and Biostatistics, Institute of Public Health, College of Medicine and Health Sciences, University of Gondar, Gondar, Ethiopia

* zephab2@gmail.com

**Data Availability Statement:** All relevant data are within the paper and its Supporting information files.

**Funding:** The author(s) received no specific funding for this work.

## Abstract

### Introduction

Hypertension is a major public health problem globally and it is a leading cause of death and disability in developing countries. This review aims to estimate the pooled prevalence of hypertension and its determinants in Ethiopia.

### Methods

A systematic literature search was conducted at the electronic databases (PubMed, Hinari, and Google Scholar) to locate potential studies. Heterogeneity between studies checked using Cochrane Q test statistics and $I^2$ test statistics and small study effect were checked using Egger's statistical test at 5% significance level. Sensitivity analysis was checked. A random-effects model was employed to estimate the pooled prevalence of hypertension and its determinants in Ethiopia.

### Results

In this review, 38 studies that are conducted in Ethiopia and fulfilled the inclusion criteria with a total number of 51,427 study participants were reviewed. The overall pooled prevalence of hypertension in the country was 21.81% (95% CI: 19.20–24.42, $I^2$ = 98.35%). The result of the review also showed that the point of prevalence was higher among males (23.21%) than females (19.62%). When we see the pervasiveness of hypertension from provincial perspective; the highest prevalence of hypertension was observed in Addis Ababa (25.35%) and the lowest was in Tigray region (15.36%). In meta-regression analysis as the mean age increases by one year, the likelihood of developing hypertension increases by a factor of 0.58 times (β = 0.58, 95% CI: 0.31–0.86, $R^2$ = 36.67). Male sex (OR = 1.29, 95% CI: 1.03–1.61, $I^2$ = 81.35%), age > 35 years (OR = 3.59, 95% CI: 2.57–5.02, $I^2$ = 93.48%), overweight and/or obese (OR = 3.34, 95% CI: 2.12–5.26, $I^2$ = 95.41%), khat chewing (OR = 1.42, 95% CI: $I^2$ = 62%), alcohol consumption (OR = 1.50, 95% CI: 1.21–1.85, $I^2$ = 64%), family history of hypertension (OR = 2.56, 95% CI: 1.64–3.99, $I^2$ = 83.28%), and

**Competing interests:** The authors have declared that no competing interests exist.

family history of diabetes mellitus (OR = 3.69, 95% CI: 1.85–7.59, $I^2$ = 89.9%) are significantly associated with hypertension.

## Conclusion

Hypertension is becoming a major public health problem in Ethiopia. Nearly two out of ten individuals who are older than 18 years living with hypertension. Sex, age, overweight and/or obese, khat chewing, alcohol consumption, and family history of hypertension and diabetes mellitus are statistically significant determinant factors for hypertension in Ethiopia. Primary attention should be given for behavioral risk factors to tackle the alarming increase of hypertension in Ethiopia.

## Introduction

Globally, more than 1.13 billion people living with hypertension, of this two-thirds living in Low and Middle-Income Countries (LMICs) [1]. In the globe, by the end of 2025 1.56 billion people will live with hypertension [2]. In Africa, 46% of adults whose age is older than 25 years and above living with hypertension [3]. The prevalence of hypertension in Africa has raised from 19.7% in 1990 to 30.8% in 2010 [4]. One in every five people live with hypertension in LMICs and studies showed that 3 out of 4 people in these countries will live with hypertension by the end of 2025 [5]. Besides, 74.7 million people living with hypertension in Sub-Saharan Africa, and it will rise to 125.5 million by the end of 2025 [6]. These trends have been strongly linked with lifestyle changes such as an increase in smoking tobacco use, excessive alcohol consumption, and physical inactivity [7, 8]. To tackle the burden of hypertension, the Pan-African Society of Cardiology (PASCAR) identified 10 action points to be implemented by African ministers to achieve a 25% decline by the end of 2025 [7, 9].

In Ethiopia, non-communicable diseases account for 39% of all causes of mortality of which, cardiovascular disease accounts for 16% [10]. On the other hand, hypertension constitutes the majority (62.3%) of all the causes of cardiovascular-related morbidity and mortality [11]. This is because high blood pressure increases the risk of life-threatening complications on vital organs like heart, blood vessels, brain, and kidney which leads to premature mortality and disability [12].

In 2015, a systematic review and meta-analysis was conducted in Ethiopia [13]; but this study did not identify the pooled effects of factors affecting the prevalence of hypertension. Besides, there are several studies published after the previous review. Therefore, this systematic review and meta-analysis gives updated pooled prevalence and factors affecting the prevalence of hypertension in Ethiopia.

Moreover, documenting the updated pooled prevalence and its determinants of hypertension will help to achieve the action plan of the Pan-African Society of Cardiology and global targets regarding hypertension. Therefore, the objective of this systematic review and meta-analysis is to synthesize updated pooled prevalence and its determinants of hypertension in Ethiopia. The finding of this review will show the trends of hypertension in Ethiopia and that can be used for health planners, policymakers, and for the community itself to curve the alarming rise of hypertension in Ethiopia.

## Methods

### Study setting and search strategies

Ethiopia is found in the horn of Africa and has nine administrative regional states and two city administrations. Potential studies were identified using electronic databases (PubMed/MEDLINE, Hinari, Google scholar) and google search. Besides, unpublished theses were also reviewed out from some research centers and library sources. The sources are reviewed limited to English language and studies published after 01/01/2000. The task of searching sources was carried out from all stated electronic databases performed on October/24/2019. All included studies defined hypertension as Systolic Blood Pressure (SBP) $\geq$ 140 mmHg and/or a Diastolic Blood Pressure (DBP) $\geq$ 90 mmHg or known hypertensive patients on treatment. The search MeSH headings were hypertension and synonyms for hypertension were used. The synonyms of hypertension are "blood pressure, high", "and blood pressures, high", "high blood pressure", and "high blood pressures". Finally, the search combination used as; "Hypertension" OR "Blood Pressure, High" OR "High Blood Pressure" OR "High Blood Pressures" OR "Blood Pressure, High" OR "Blood Pressures, High" AND Ethiopia (S1 Table).

### Eligibility criteria

We used CoCoPop (Condition, Context, and Population) approach for prevalence studies to declare inclusion and exclusion criteria.

### Inclusion criteria and exclusion criteria

Studies conducted on the prevalence and/or associated factors of hypertension in Ethiopia were included. Besides, all full-text articles written in English language (with response rate > 85%), with participants older than 18 years and published after January 01/2000 are included for this review. Studies conducted on pregnancy-induced hypertension, for the reason that has no prevalence report on hypertension, and hypertension prevalence reports on other comorbidities excluded for this review.

### Measurement of the outcome variable

The primary outcome of interest for this review was to estimate the pooled prevalence of hypertension and its determinants. Potentials of extracted factors from each study considered as an independent factor for hypertension.

### Study selection and data collection

All the studies reviewed through different electronic databases were combined, exported, and managed using Endnote version X9.2 (Thomson Reuters, Philadelphia, PA, USA) software. All duplicate studies were removed and full-text studies downloaded using Endnote software and manually. The eligibility of each study was completely assessed independently by two reviewers (SA. &YA.). Exaggerated differences in the results of the two reviewers narrowed through discussion and other reviewer members (ST. & DA.).

### Assessment of the quality of the individual studies

The quality of the studies assessed using the validated modified version of a quality assessment tool for prevalence studies [14]. Two reviewers (SA. & YA.) were independently assessed to check the quality of the included studies. The problem of subjectivities between the two reviewers was solved through discussion and other review teams (ST. & DA.). The quality

assessment tool has nine-questions. Based on the score of the quality assessment tool the highest score had the minimum risk of bias. Overall scores range from (0–3), (4–6), and (7–9), which are declared low, moderate, and high risk of bias respectively [14].

### Data extraction and management

All-important parameters extracted from each study were reviewed by two authors (SA. &YA.) independently using Microsoft Excel. The discrepancies between the two authors managed through discussion and/or the other authors (ST. & DA.). The data extraction format was prepared using the assistance of the Joanna Briggs Institute (JBI) data extraction tool for prevalence studies. For each study, authors, years of publication, study design, sample size, the prevalence of hypertension with their standard error, and determinant factors effect size with their standard error were extracted.

### Statistical analysis

The extracted data were exported to STATA/MP version 16.0 software for analysis. The pooled prevalence of hypertension and its determinants analyzed by the random effects model using DerSimonian-Laird model weight [15]. Heterogeneity in meta-analysis is mostly inevitable due to differences in study quality, its sample size, method, and different outcome measurements across studies [16, 17]. Statistically, significant heterogeneity was checked by Cochrane Q-test and $I^2$ statistics [18]. To minimize the variance of estimated points between primary studies, a subgroup analysis was carried out in reference to the regions, age categories, and residence. Besides, a sensitivity analysis was also conducted to determine the influence of single studies on the pooled estimates. Univariate meta-regression conducted using year of publication, the mean age of the respondent from primary studies, sample size, and region using random effects model. Publication bias (small study effect) checked using graphically and Egger's statistical test [19]. Statistically significant Egger's test (P-value < 0.05) indicates that the presence of a small study effect and handled by non-parametric trim and fill analysis using the random effects model [20].

## Results

### Study selection and identification

Of the 784 studies reviewed, 336 were excluded, because they were duplications. By reading their titles and abstracts, 406 studies excluded as they were irrelevant for this review. Again, five studies excluded, because of the outcome not reported, inadequate sample size, and lack of full text. Finally, 38 potential studies have been included for qualitative and quantitative synthesis influences as summarized in the PRISMA flow diagram [21] (Fig 1).

### Characteristics of included studies

Among the included studies, 20 (52.60%) studies published after 2016. All the included studies were cross-sectional surveys, of which 27 community based, six health facility-based, and five studies were institutional-based (Schools, College, Bank. . .). Overall, a total number of 51,427 study participants who are older than 18 years included for this review. The minimum and maximum sample sizes were 306 and 9788 respectively [22, 23]. A minimum of (7.47%) and maximum of (41.90%) prevalence of hypertension were reported from the studies conducted in the Oromia region [24, 25]. Five regions and two city administrations (Addis Ababa and Dire Dawa) were represented for this review. Seven from Amhara Region [26–32], eight from Oromia Region [24, 25, 33–38], six from South National and Nationalities of People's Region

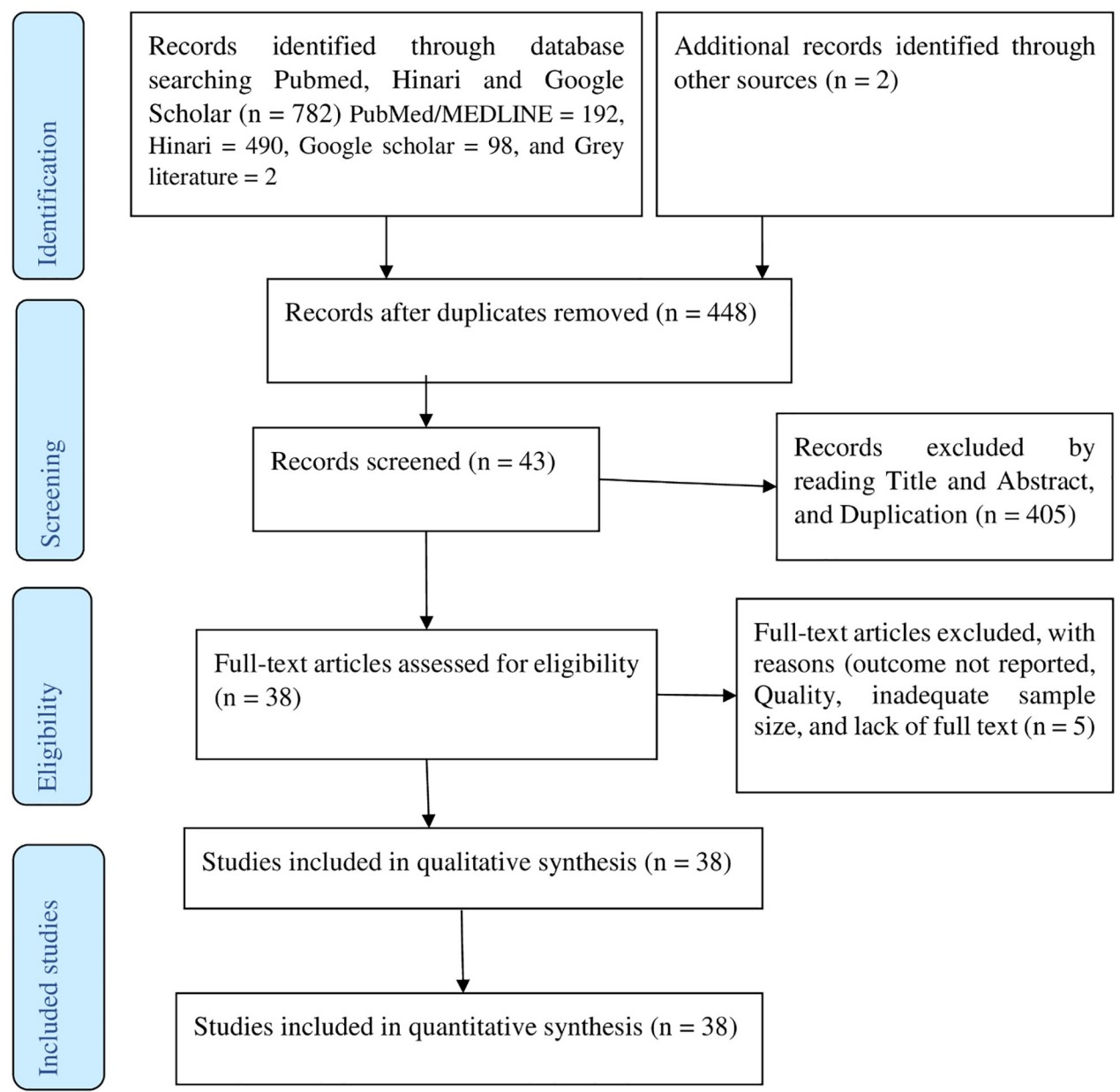

**Fig 1. PRISMA flow diagram of article selection for systematic review and meta-analysis of the prevalence of hypertension and its determinants in Ethiopia.**

(SNNPR) [22, 39–43], four from Tigray Region [44–47], three from Somali Region [48–50], eight from Addis Ababa [51–58], two from Dire Dawa [37] and one national study in Ethiopia [23] were included. No studies reviewed from Gambela, Afar, Benishangul Gumez, and Harari Regional states of Ethiopia (Table 1).

## The pooled prevalence of hypertension in Ethiopia

In random effects model, the pooled prevalence of hypertension in Ethiopia was 21.81 (95% CI = 19.20–24.42); significant heterogeneity observed among studies ($I^2$ = 98.4, P-

**Table 1. Characteristics of the included studies and their prevalence of hypertension in Ethiopia, 2019.**

| S. No | Author | Publication year | Region | Sample size | Response rate (%) | Prevalence of hypertension | Quality score |
|---|---|---|---|---|---|---|---|
| 1 | Zekewos et al. [42] | 2019 | SNNPR | 425 | - | 21.80 | 1 |
| 2 | Kiber et al. [30] | 2019 | Amhara | 456 | 95.6 | 12.50 | 2 |
| 3 | Shukuri et al. [25] | 2019 | Oromia | 401 | 96 | 41.90 | 1 |
| 4 | Abebe et al. [56] | 2019 | Addis Ababa | 487 | 100 | 34.70 | 0 |
| 5 | Roba et al. [59] | 2019 | Dire Dawa | 872 | 96.5 | 24.40 | 3 |
| 6 | Belachew et al. [27] | 2018 | Amhara | 308 | 100 | 27.30 | 0 |
| 7 | Gebreyes et al. [23] | 2018 | National | 9788 | 95.4 | 18.05 | 0 |
| 8 | Bayray et al. [46] | 2018 | Tigray | 1523 | 99.7 | 15.90 | 0 |
| 9 | Tesfaye et al. [38] | 2018 | Oromia | 648 | 97 | 14.2 | 0 |
| 10 | Esaiyas et al. [40] | 2018 | SNNPR | 620 | 99.6 | 19.70 | 0 |
| 11 | Bekele et al. [51] | 2018 | Addis Ababa | 758 | 100 | 15.90 | 0 |
| 12 | Asfaw et al. [41] | 2018 | SNNPR | 524 | 99.8 | 30.00 | 0 |
| 13 | Mara et al. [39] | 2018 | SNNPR | 346 | 97.4 | 23.00 | 0 |
| 14 | Neba et al. [50] | 2017 | Somali | 548 | 100 | 21.90 | 0 |
| 15 | Demisse et al. [28] | 2017 | Amhara | 3057 | 94.8 | 27.40 | 0 |
| 16 | Asresahegn et al. [48] | 2017 | Somali | 487 | 98.9 | 28.30 | 0 |
| 17 | Birhanu Tolera [57] | 2017 | Addis Ababa | 401 | 98.5 | 14.00 | 1 |
| 18 | Seifu et al. [49] | 2017 | Somali | 330 | 100 | 13.30 | 1 |
| 19 | Gebrihet et al. [45] | 2017 | Tigray | 521 | 96 | 16.50 | 0 |
| 20 | Fikadu et al. [52] | 2016 | Addis Ababa | 1866 | 100 | 21.00 | 0 |
| 21 | Tadele et al. [22] | 2016 | SNNPR | 306 | 95.9 | 27.80 | 3 |
| 22 | Abdissa et al. [54] | 2015 | Addis Ababa | 2716 | 100 | 24.90 | 1 |
| 23 | Anteneh et al. [32] | 2015 | Amhara | 678 | 99.6 | 25.10 | 1 |
| 24 | Asresahegn et al. [34] | 2015 | Oromia | 830 | 100 | 36.40 | 3 |
| 25 | Angaw et al. [55] | 2015 | Addis Ababa | 629 | 96 | 27.30 | 0 |
| 26 | Birlew et al. [24] | 2015 | Oromia | 4055 | 90.7 | 7.47 | 2 |
| 27 | Abebe et al. [26] | 2015 | Amhara | 2141 | 97.3 | 27.90 | 1 |
| 28 | Bissa et al. [35] | 2014 | Oromia | 701 | 96.02 | 21.30 | 1 |
| 29 | Zikru et al. [47] | 2014 | Tigray | 709 | 99.7 | 11.00 | 3 |
| 30 | Mengistu et al. [44] | 2014 | Tigray | 1183 | 100 | 18.10 | 1 |
| 31 | Tadesse et al. [31] | 2014 | Amhara | 610 | 100 | 7.70 | 2 |
| 32 | Helelo et al. [43] | 2014 | SNNPR | 518 | 96.6 | 22.40 | 2 |
| 33 | Gudina et al.[36] | 2014 | Oromia | 396 | 93.8 | 16.92 | 2 |
| 34 | Gudina et al. [33] | 2013 | Oromia | 734 | 100 | 13.20 | 0 |
| 35 | Nshisso et al. [53] | 2012 | Addis Ababa | 2153 | 100 | 19.10 | 1 |
| 36 | Awoke et al. [29] | 2012 | Amhara | 679 | 97.6 | 28.30 | 0 |
| 37 | Muluneh et al. [37] | 2012 | Oromia | 3223 | - | 9.30 | 2 |
| 38 | Tesfaye et al. [58] | 2009 | Addis Ababa | 648 | 93.2 | 14.20 | 1 |

value < 0.001). The highest weight among studies observed from the studies conducted by Muluneh et al. [37], Gebreyes et al. [23], and Birlew et al. [24] (Fig 2). Among 23 studies in the random effects model, the pooled prevalence of hypertension among males were 23.21 (95% CI:18.86–27.57) (Fig 3) with statistically significant heterogeneity ($I^2 = 97.5\%$, P-value < 0.001). Besides, the overall pooled prevalence of hypertension among females were 19.62 (95% CI: 16.26–22.97) (Fig 4); heterogeneity ($I^2 = 96.08\%$, P-value <0.001). Egger's

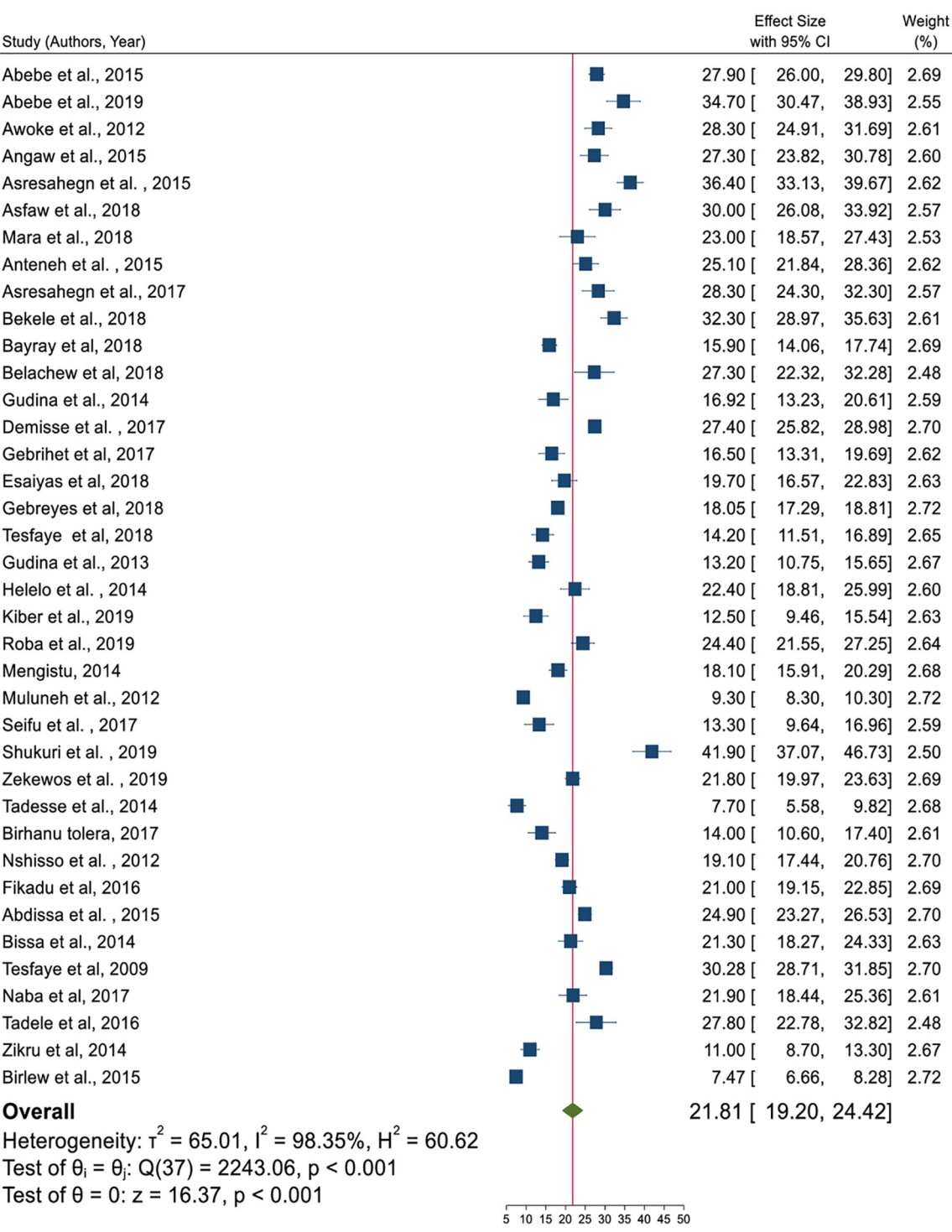

**Fig 2. Pooled prevalence of hypertension age greater than 18 years in Ethiopia.**

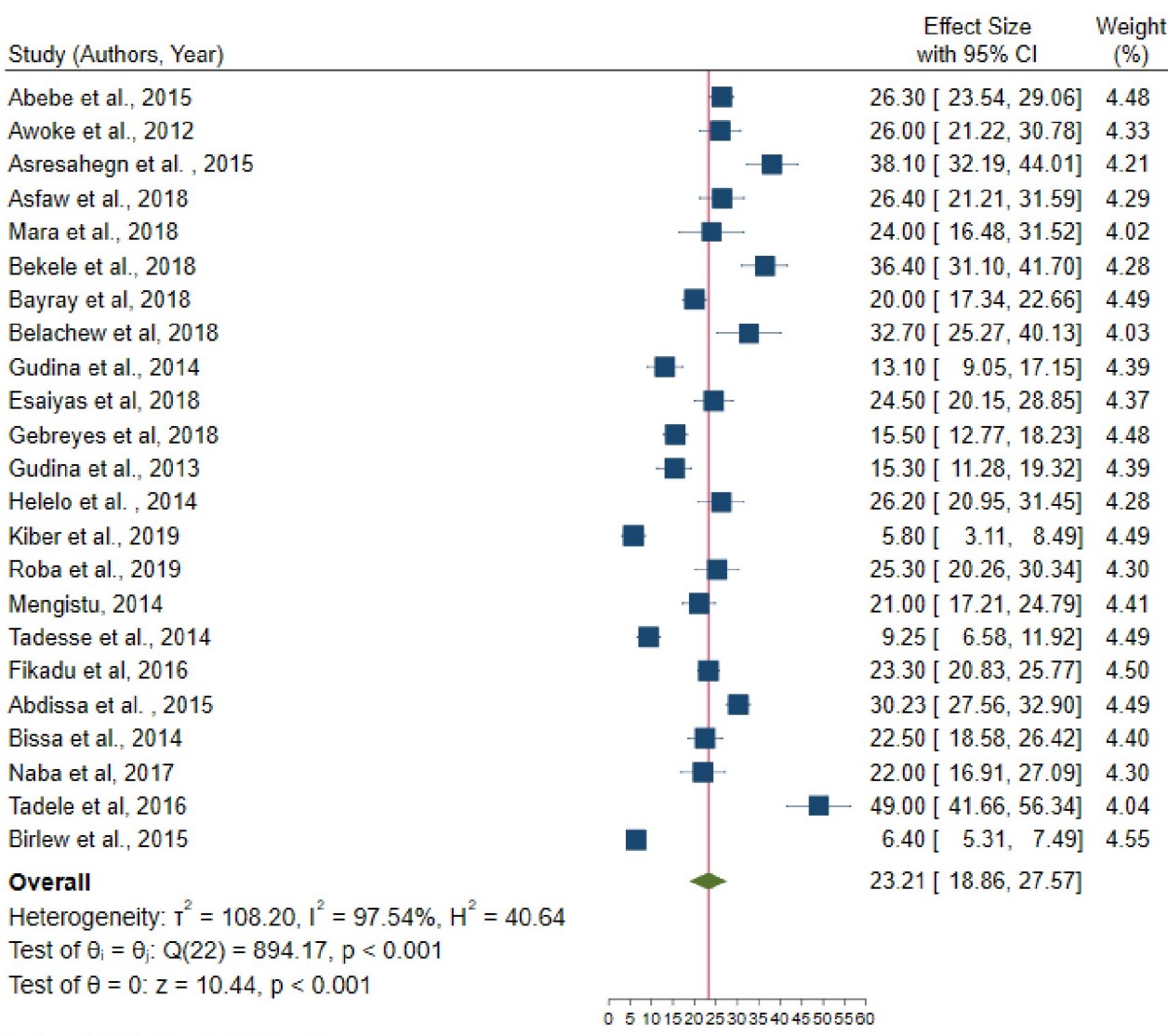

**Fig 3. Pooled prevalence of hypertension among males in Ethiopia, 2019.**

statistical test evidenced that has no publication bias among the included studies (β = -0.615, P-value = 0.91).

## Handling heterogeneity

Significant heterogeneity observed from random effects model pooled estimate. To handle this heterogeneity sensitivity analysis, subgroup analysis, and meta-regression analysis were performed.

## Sensitivity analysis

From the random effects model, there are no studies that excessively influence the overall pooled estimate of hypertension (S1 Fig).

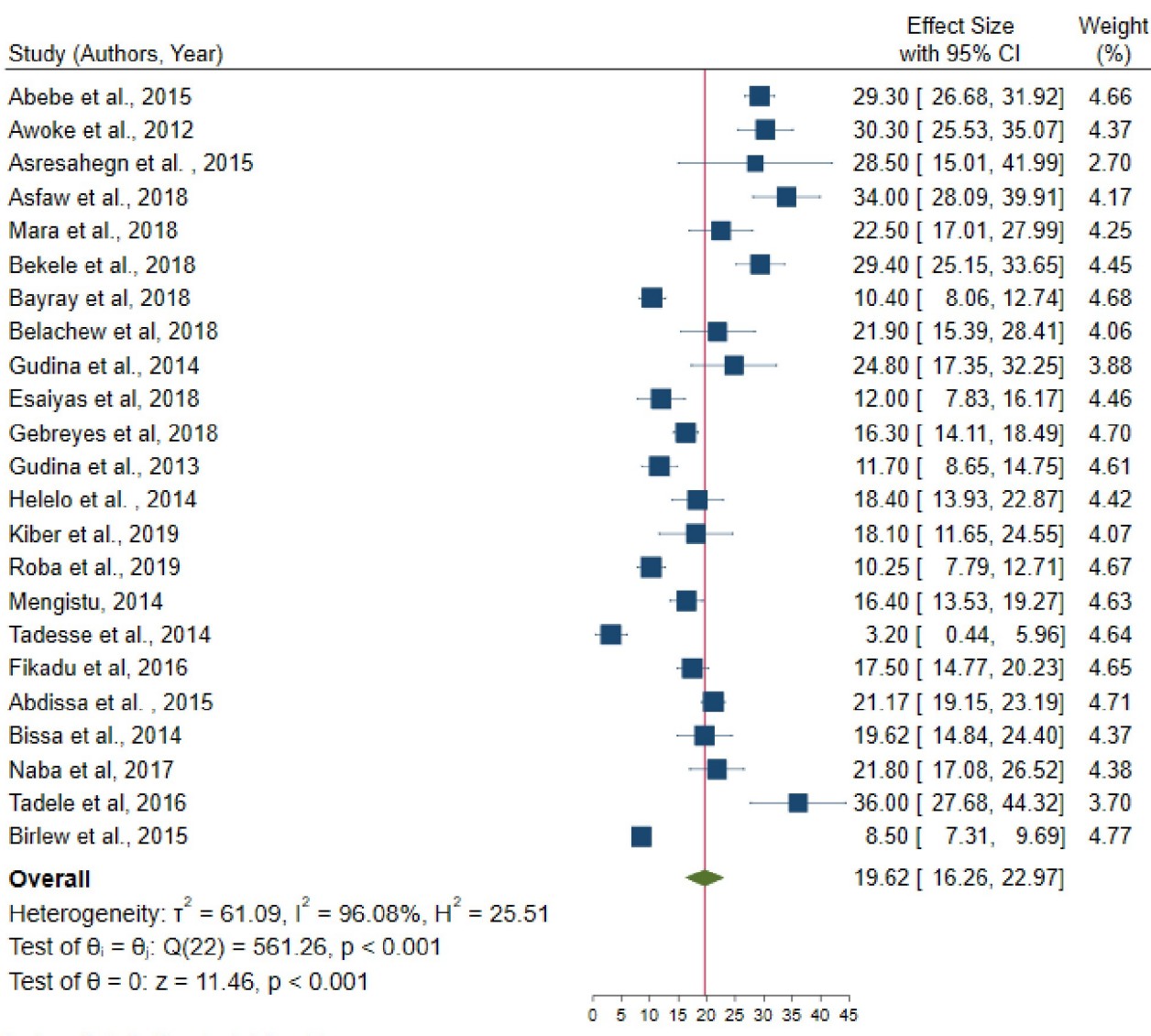

**Fig 4. Pooled prevalence of hypertension among females in Ethiopia, 2019.**

## Subgroup analysis

Even though subgroup analysis was carried out across the administrative regions of the country, age category, and residence as the source of heterogeneity was not handled. In the subgroup analysis, the highest prevalence of hypertension observed in Addis Ababa (25.35%) followed by Southern Nations Nationalities and People's Region (23.83%); whereas the lowest prevalence was in Tigray regional state of Ethiopia (15.36%). The pooled prevalence of hypertension (27%) was higher in the age category which is older than 35 years. Also, the highest prevalence of hypertension was observed in urban inhabitants (22.85%) (Table 2).

**Table 2. Sub-group pooled prevalence of hypertension in Ethiopia, 2019 (n = 38).**

| Variables | | Included studies | Sample size | Prevalence (95%CI) | Heterogeneity (I², p-value) |
|---|---|---|---|---|---|
| By region | Tigray | 4 | 3936 | 15.36 (12.33–18.39) | 85.4%, < 0.001 |
| | Amhara | 7 | 7929 | 22.27 (15.44–29.11) | 98.1%, < 0.001 |
| | Oromia | 8 | 10988 | 19.83 (14.09–25.28) | 98.7%, < 0.001 |
| | SNNPR | 6 | 2739 | 23.83 (20.93–26.72) | 77.0%, < 0.001 |
| | Addis Ababa | 8 | 9658 | 25.35 (21.25–29.45) | 96.3%, < 0.001 |
| | Somali | 3 | 1365 | 21.14 (12.86–29.42) | 93.3%, < 0.001 |
| By age category | > 18 years | 25 | 38360 | 19.92 (24.28–29.56) | 98.4%, < 0.001 |
| | > 25 years | 7 | 8304 | 24.37 (19.84–28.89) | 95.4%, < 0.001 |
| | > 30 years | 2 | 1196 | 23.86 (21.22–26.49) | 15.9%, 0.275 |
| | > 35years | 3 | 2888 | 26.92 (24.28–29.56) | 53.7%, 0.115 |
| By residence | Rural | 5 | 10814 | 18.45 (12.41–24.48) | 99.03, < 0.001 |
| | Urban | 28 | 26554 | 22.85 (20.34–25.36) | 95.91, < 0.001 |
| | Both urban and rural | 5 | 14059 | 18.45 (12.41–24.48) | 98.19, < 0.001 |

## Meta-regression

Univariate meta-regression analysis revealed that the mean age and region were statistically significant with hypertension. As the mean age increased by one year, the likelihood of developing hypertension increases by a factor of 0.58 times (β = 0.58, 95% CI: 0.31–0.86); with a total proportion of hypertension explained by the covariate mean age by 36.67% (adjusted $R^2$ = 36.67). The linear relationship between mean age and hypertension was presented as shown in Fig 5 below. Besides, the pooled prevalence of hypertension was higher in the capital city of

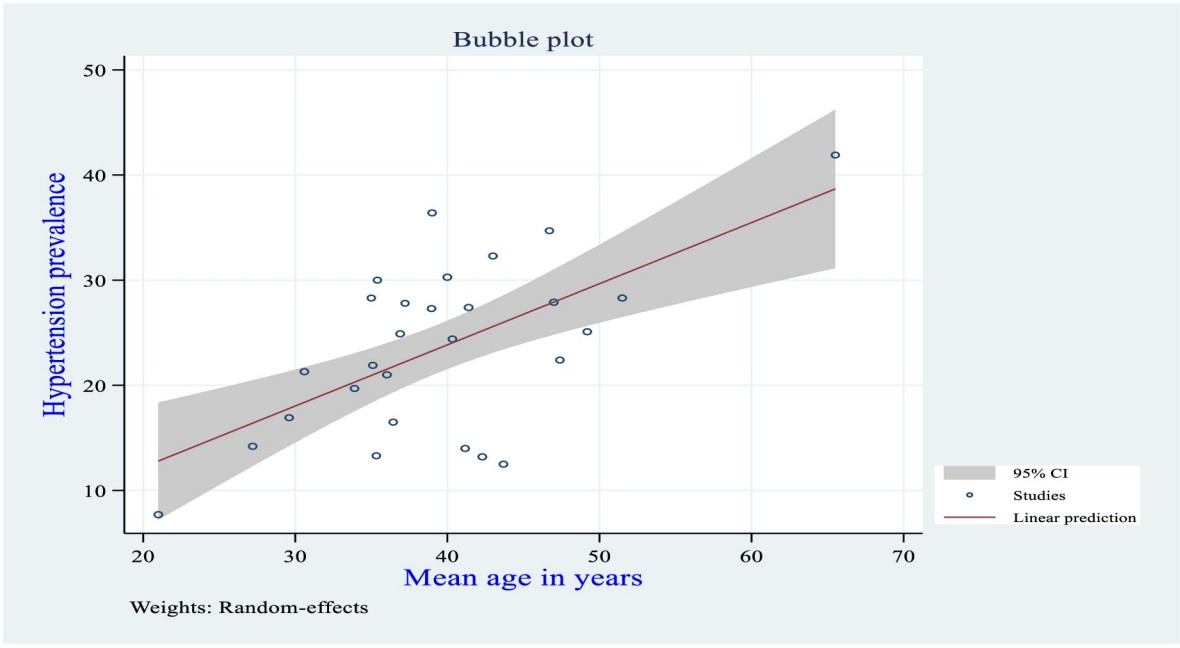

**Fig 5. The relationship between mean age and hypertension in the Ethiopian population, 2019.**

**Table 3. Univariate meta-regression analysis results for the prevalence of hypertension in Ethiopia, 2019.**

| Study level variables | | Adjusted $R^2$ | Standard error | Coefficients (95% CI) |
|---|---|---|---|---|
| Mean age | | 36.67 | 0.14 | 0.58 (0.31–0.86) * |
| Publication year | | 00 | 0.57 | 0.58(-0.54–1.69) |
| Sample size | | 00 | 0.0008 | 0.00072 (-0.0023–0.0009) |
| Regions | Tigray | 1 | 1 | 1 |
| | Amhara | | 4.59 | 6.88 (-2.13–15.89) |
| | Addis Ababa | 20 | 4.48 | 10.01(1.22–18.80) * |
| | Oromia | | 4.49 | 4.39 (-4.40–13.19) |
| | SNNPR | | 4.76 | 8.67 (-0.65–18.00) |
| | Somali | | 5.64 | 5.77 (-5.30–16.54) |
| | Dire Dawa | | 8.21 | 9.03 (-7.07–25.12) |

NB:

* = Statistically significant at 5% level, CI = Confidence Interval.

Addis Ababa, Ethiopia as compared to Tigray regional state of Ethiopia (β = 10.01, 95% CI: 1.22–18.80) (Table 3).

## Factors associated with hypertension

As summarized in Table 4, sex, age, Body Mass Index (BMI), chat chewing, alcohol consumption, and family history of hypertension and diabetes mellitus were statistically significant factors for hypertension.

**Table 4. Summary of the pooled effects of factors associated with hypertension in Ethiopia, 2019.**

| Variables | | OR (95% CI) | Heterogeneity ($I^2$, P-value) | Egger's P-value | Total studies | Sample size |
|---|---|---|---|---|---|---|
| Sex | Female | 1 | | | | |
| | Male | 1.29 (1.03–1.61) * | 81.35%, < 0.001 | 0.544 | 15 | 19957 |
| Age | < 35 years | 1 | 1 | | | |
| | > 35 years | 3.59 (2.57–5.02) * | 93.48%, < 0.001 | 0.487 | 15 | 27365 |
| BMI | Normal | 1 | | | | |
| | Underweight | 0.68 (0.30–1.56) | 94.00%, < 0.001 | 0.229 | 16 | |
| | Overweight and /or obese | 3.34 (2.12–5.26) * | 95.41%, < 0.001 | 0.176 | 18 | 13383 |
| Khat chewing | No | 1 | 1 | | | |
| | Yes | 1.42 (1.10–1.85) * | 62.2%, 0.005 | 0.267 | 10 | 8687 |
| Smoking | No | 1 | 1 | | | |
| | Yes | 1.55 (1.00–2.38) | 67.56%, 0.002 | 0.873 | 10 | 9556 |
| Alcohol drinking | No | 1 | 1 | | | |
| | Yes | 1.50 (1.21–1.85) * | 64.0%, 0.001 | 0.005 | 14 | 12988 |
| Physical activity | Active | 1 | 1 | | | |
| | Inactive | 1.24 (0.83–1.85) | 91.28%, < 0.001 | 0.0002 | 15 | |
| Family history of HTN | No | 1 | 1 | | | |
| | Yes | 2.56 (1.64–3.99) * | 83.28%, < 0.001 | 0.016 | 11 | 5918 |
| Family history of DM | No | 1 | 1 | | | |
| | Yes | 3.69 (1.85–7.59) * | 89.93%, < 0.001 | 0.4707 | 9 | 14660 |

NB:

* = Statistically significant at 5% level, OR = Odds Ratio, CI = Confidence Interval.

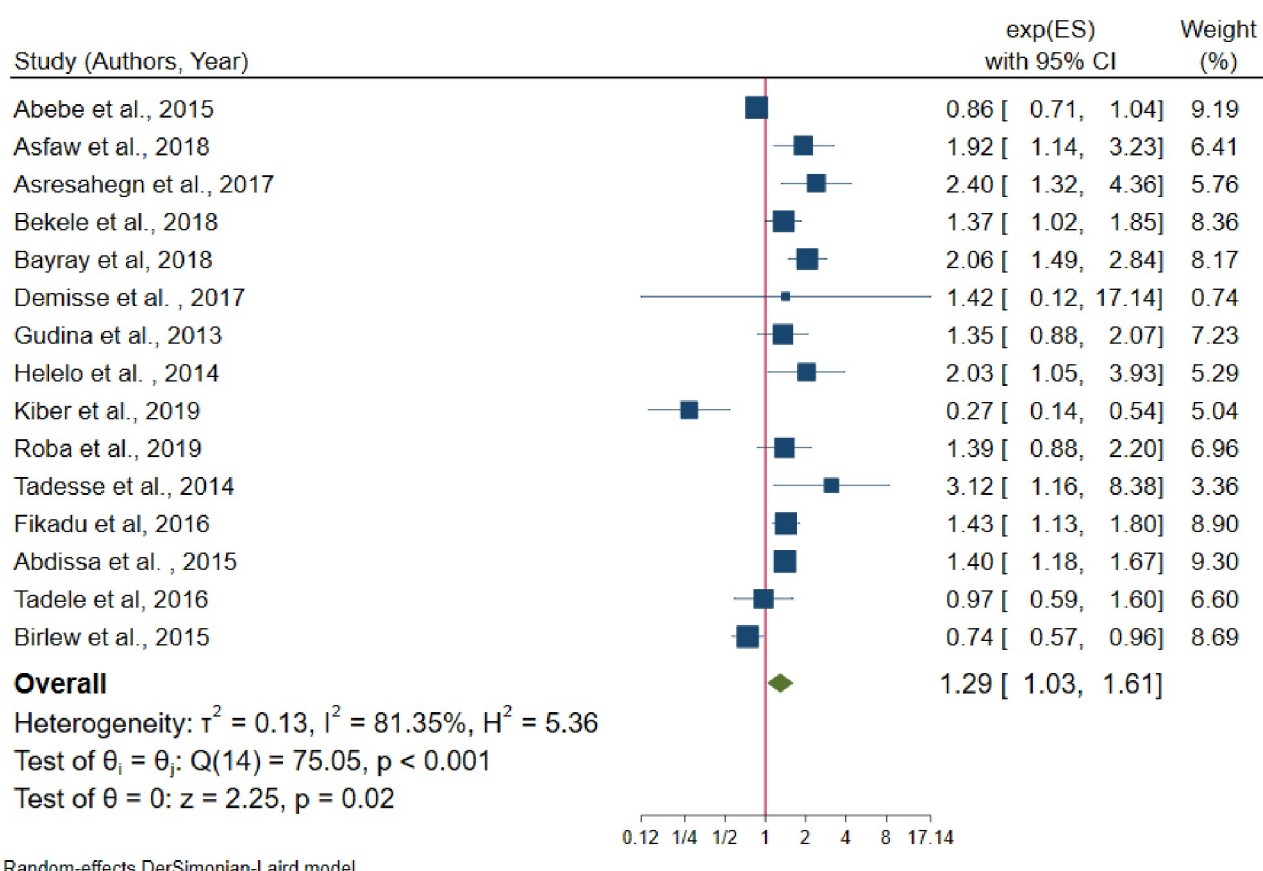

**Fig 6. The association between males and hypertension.**

Fifteen [22, 24, 26, 28, 30, 31, 33, 41, 43, 46, 48, 51, 52, 54, 59] studies were included to identify the association between sex and hypertension. Five of these studies [22, 26, 28, 33, 59] had no statistically significant association between sex and hypertension. From random effects model estimate, the pooled odds of developing hypertension among males were 29% more likely to develop hypertension than females (OR = 1.29, 95% CI: 1.03–1.61); with statistically significant heterogeneity between studies ($I^2$ = 81.3%, P-value < 0.001) (Fig 6). Egger's test indicates that no small study effect (P-value = 0.544) and in random effects model there was no single study that excessively influences the pooled estimate of hypertension (S2 Fig).

The pooled effect of age has a significant association with hypertension. From fifteen [23, 24, 28, 30, 35, 40, 41, 46, 48, 50, 51, 52, 54–56] studies only one [24] study had no significant association between age and hypertension. The pooled odds of developing hypertension among individuals older than 35 years was 3.59 times higher than age younger than 35 years (OR = 3.59, 95% CI: 2.57–5.02) (Fig 7); with statistically significant heterogeneity among studies ($I^2$ = 93.5%, P-value < 0.001). There is no small study effect (P-value = 0.485) and in random effects model, there was no single study excessively influence the pooled estimate of effect size (S3 Fig).

A total of eighteen [22, 25, 27–29, 31, 32, 35, 40, 43, 45–47, 49–51, 55, 56] studies included to estimate the association between BMI and hypertension. The results of the test statistics indicate that significant heterogeneity was observed between studies ($I^2$ = 95.41%, P-value < 0.001). Egger's test evidenced that there was no publication bias (P-value = 0.176).

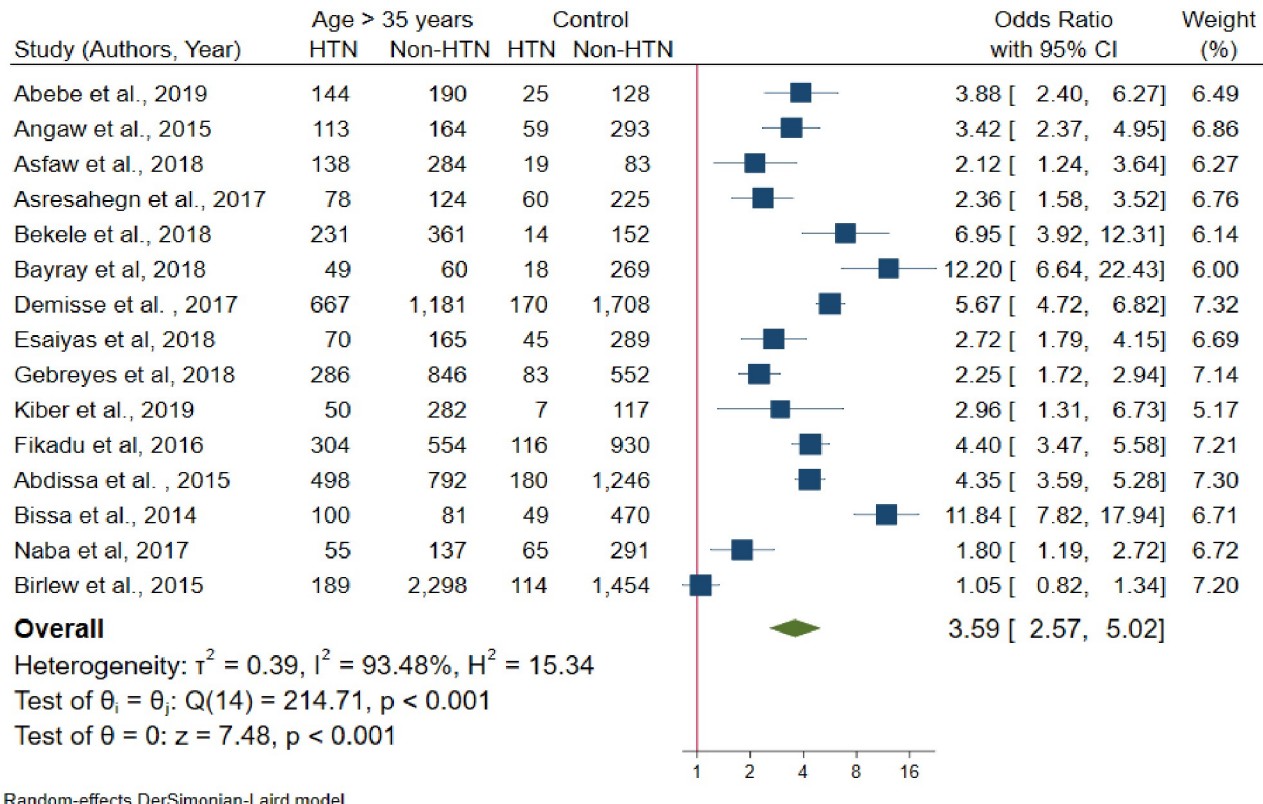

**Fig 7. Forest plot for the association between age and hypertension.**

Again, from random effects model, no individual studies excessively influence the pooled estimate of the effect size (S4 Fig). From the random effects model pooled estimate, the likelihood of developing hypertension among overweight and/or obese individuals was 3.34 times higher than the normal-weight individuals (OR = 3.34, 95% CI: 2.12–5.26) (Fig 8).

The pooled effects between khat chewing and hypertension was assessed using ten studies [22, 24, 33, 35, 36, 46, 48, 49, 55, 57]. Among the included studies, six [22, 24, 33, 36, 38, 48] of them reported that khat chewing has not a statistically significant association with hypertension. Based on Egger's test there was no publication bias (P-value = 0.498). Besides, from random effects model there was no single study that excessively influences the pooled effect size (S5 Fig). Khat chewers have 42% more likelihood to develop hypertension than non-khat chewers (OR = 1.42, 95% CI: 1.10–1.85) (Fig 9), with moderate heterogeneity ($I^2$ = 62.2%, P-value = 0.005).

The association between alcohol consumption and hypertension was assessed using 14 studies [22, 24, 26, 27, 30, 34–36, 38, 55–57, 59]. Moderate heterogeneity was also observed from the random effects model ($I^2$ = 64.04%) and there is no evidence of a single study that affects the pooled effects size in the sensitivity analysis (S6 Fig). Egger's test evidenced that small study effect (P-value = 0.001). After non-parametric trim and fill analysis (Fig 10), alcohol consumption had a negative effect on hypertension. From the random-effects trim and fill analysis, alcohol drinkers were more likely to develop hypertension by half as compared to non-drinkers (OR = 1.50, 95% CI: 1.21–1.85).

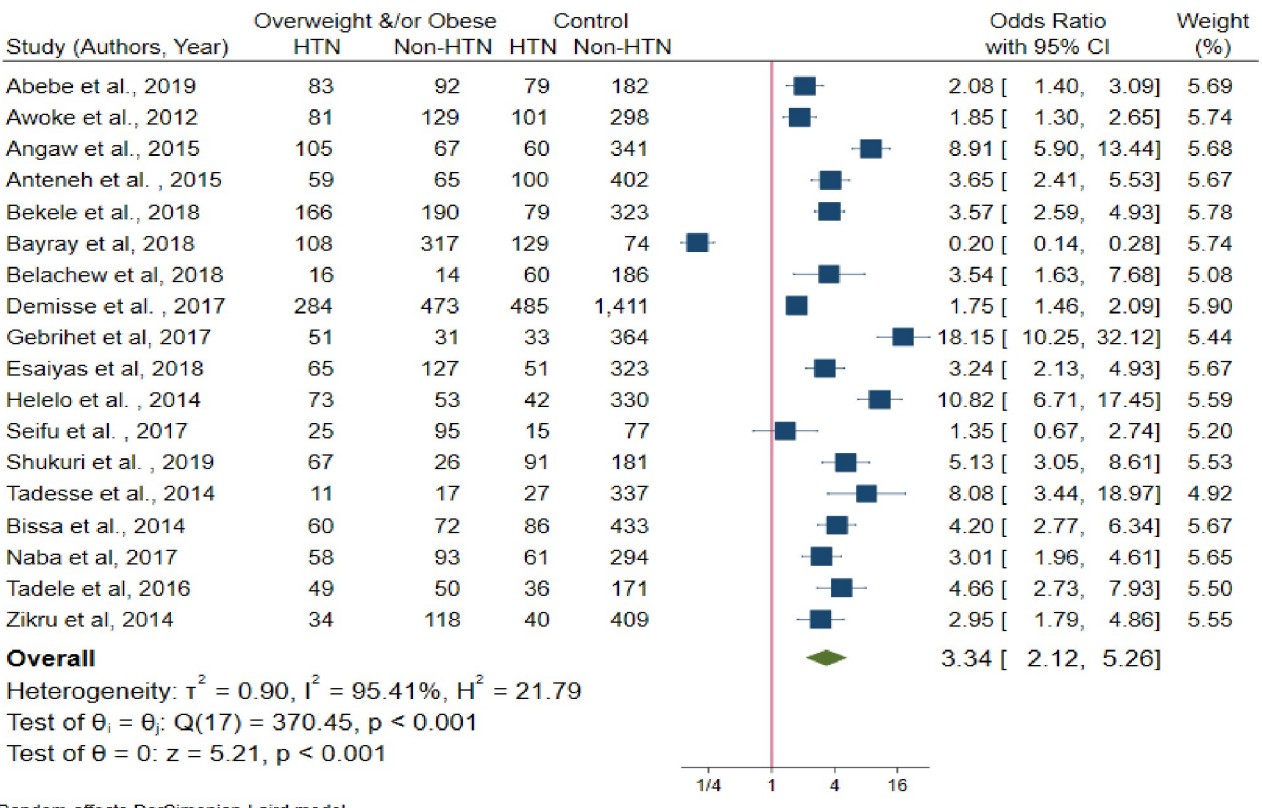

**Fig 8. The association between body mass index and hypertension.**

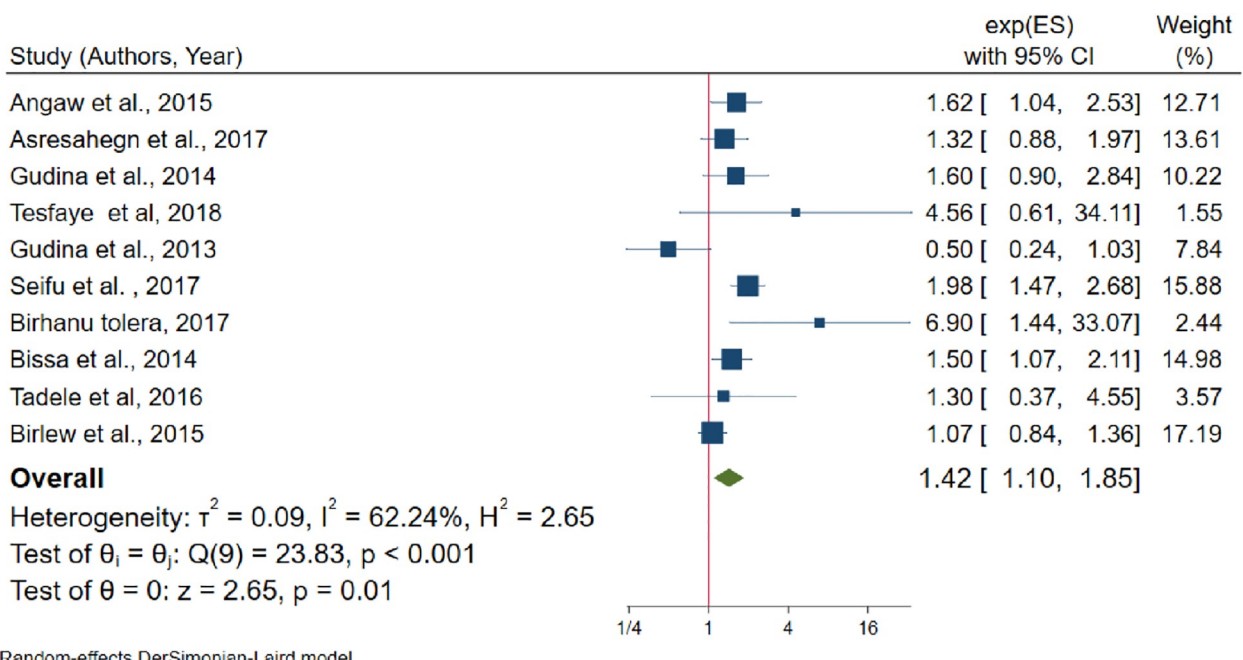

**Fig 9. The association between khat chewing and hypertension.**

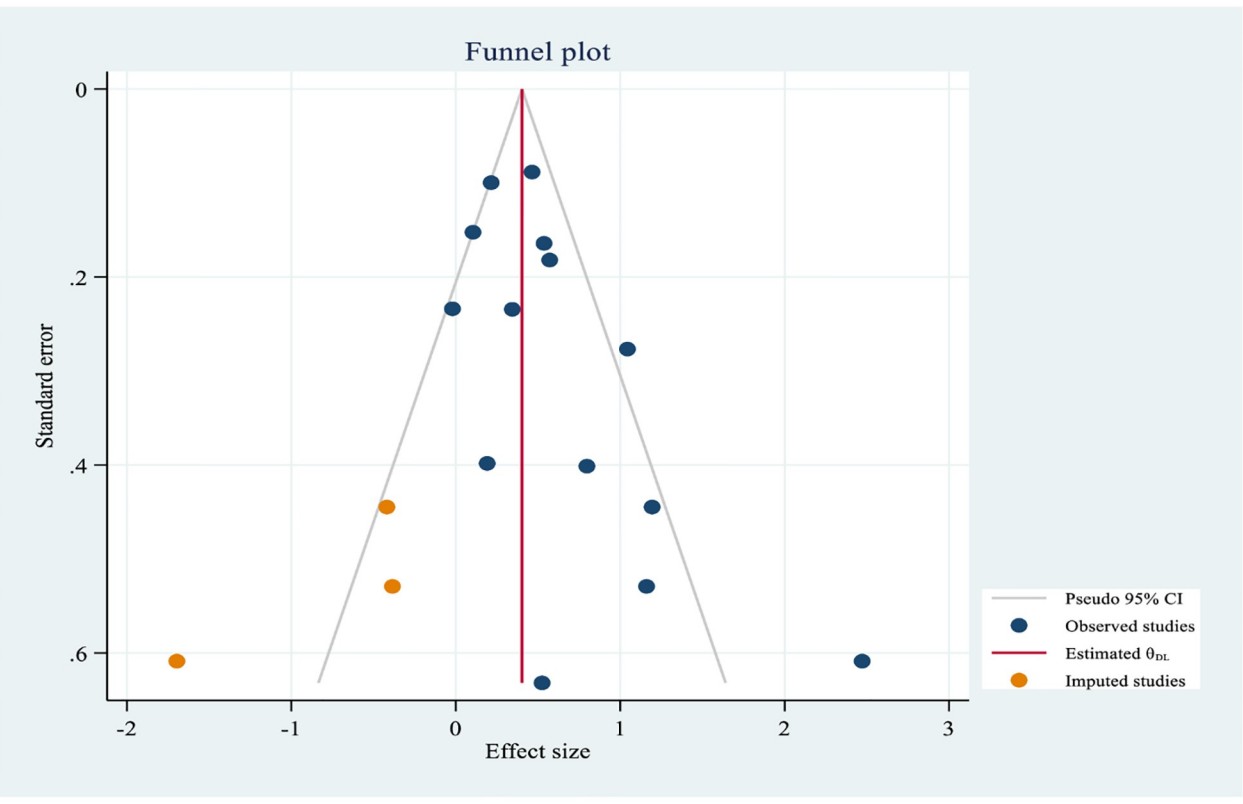

**Fig 10. Trim and fill analysis funnel plot for alcohol consumption.**

A total of fifteen studies [22, 25, 26, 28, 29, 32, 33, 35, 41, 43, 45, 48, 49, 56, 59] were included to determine the association between physical activity and hypertension; of them, four studies had no statistically significant association with hypertension. From random effects model estimate, significant heterogeneity observed ($I^2$ = 91.3%, P-value < 0.001). Egger's test indicates that evidence of publication bias (P-value = 0.002). After non-parametric trim and fill analysis, physical exercise and hypertension has no significant association (OR = 1.24, 95% CI: 0.83–1.85).

As the results of eleven studies [22, 25, 29, 30, 33, 34, 43, 48–50, 55], family history of hypertension and hypertension had statistically significant association. The random effects model evidenced that statistically significant heterogeneity across studies ($I^2$ = 83.3%, P-value < 0.001). From the sensitivity analysis random effects model estimate there is no single study that excessively influences pooled effect size (S7 Fig). Egger's test showed that the presence of a small study effect (P-value = 0.016). After non-parametric trim and fill analysis pooled estimate (Fig 11), the pooled odds of developing hypertension among individuals who had a family history of hypertension were 2.56 times higher than their counterparts (OR = 2.56, 95% CI:1.64–3.99).

Furthermore, the association between the family of diabetes mellitus and hypertension was identified using nine studies [22, 23, 29, 32, 33, 48, 55, 56, 59]; among them, four studies [22, 23, 48, 59] showed that there is no statistically significant association between family history of diabetes mellitus and hypertension. The random effects model estimate showed that statistically significant heterogeneity between studies ($I^2$ = 89.9%, P-value < 0.001) and Egger's test

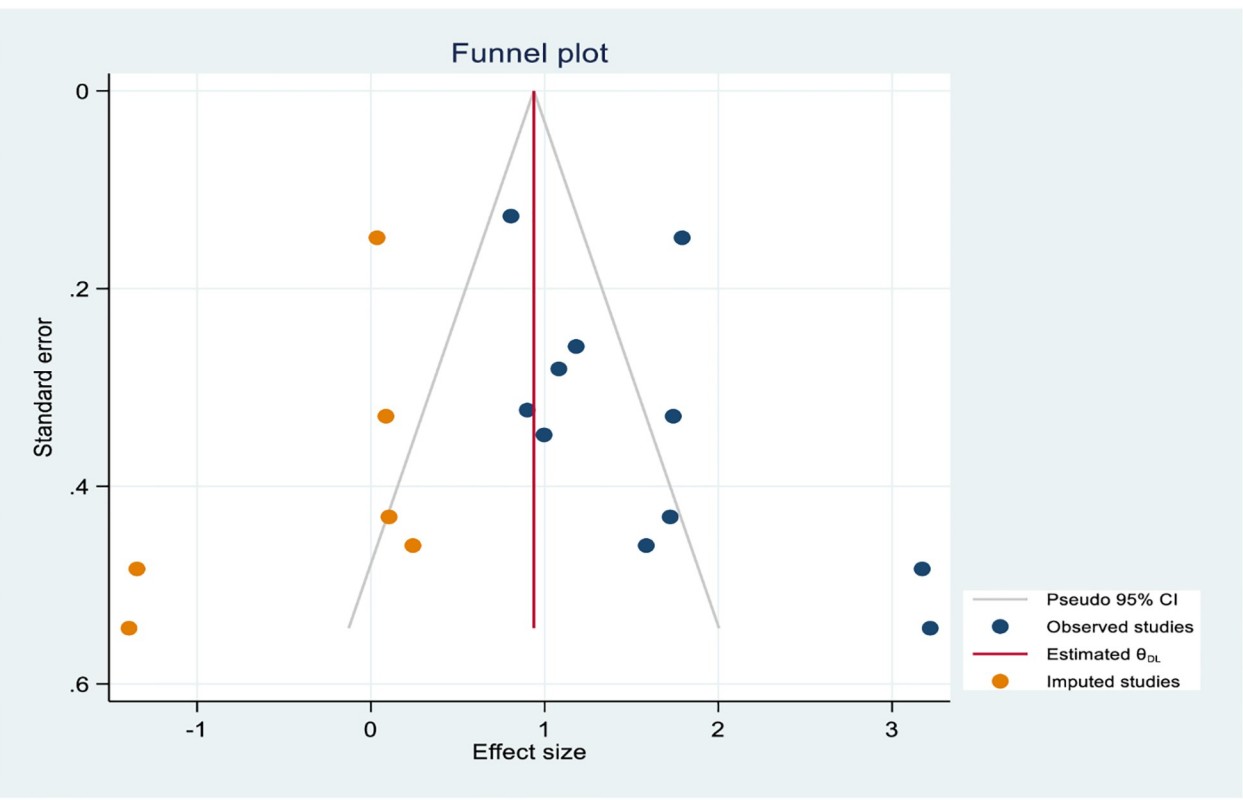

**Fig 11. Trim and fill analysis funnel plot for a family history of hypertension.**

showed that there is no publication bias (P-value = 0.47). From random effects model sensitivity analysis, there is no single study that excessively affects the pooled effect size (S8 Fig). Form random effects model estimate individuals who had a family history of diabetes mellitus are 3.69 times more likely to develop hypertension than the reference category (OR = 3.69, 95% CI: 1.85–7.59) (Fig 12) (Table 4).

## Discussion

Non-communicable diseases are becoming a double burden of public health problem in developing countries [60]; besides hypertension prevalence is rising in developing countries in contrast to developed nations [61]. This systematic review and meta-analysis will give the update pooled estimates of hypertension in Ethiopia which gives invaluable information to policymakers, health planners, and the community itself.

This systematic review and meta-analysis revealed that the pooled prevalence of hypertension in Ethiopia was 21.81% (95% CI: 19.20–24.42), which was consistent with a study conducted in rural communities of Sub-Saharan Africa (22%), Kenya (22.8%), and a meta-analysis from Vietnam (21.1%) [62–64]. However, the finding of this meta-analysis was lower than the previous meta-analysis reports in LMICs (32.3%), among adults in Africa (57.0%), a meta-analysis study on undiagnosed hypertension in Sub-Saharan Africa (30%), Nigeria (28.9%), India (29.8%), Pakistan (26.34%), and a study in Nepal (25.1%) [65–71]. The prevalence of hypertension in this review was higher than a study conducted a previous systematic review in Ethiopia and a study conducted in Ghana [13, 72]. The possible reason for this discrepancy

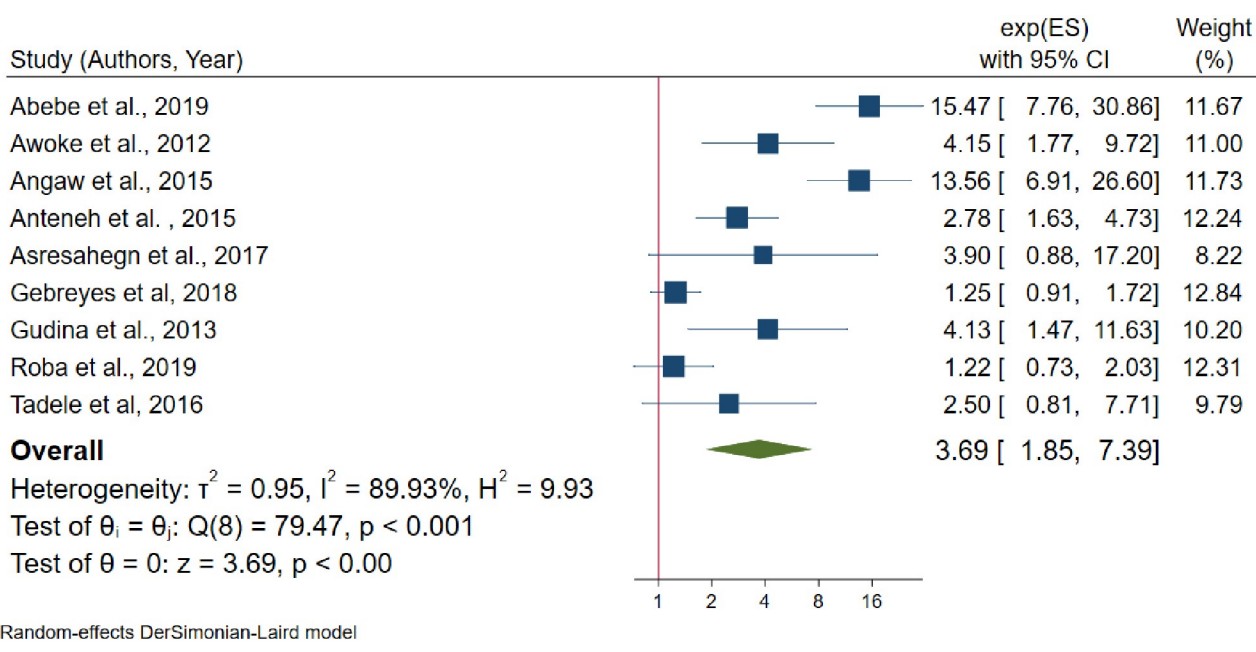

**Fig 12. The association between family history of diabetes mellitus and hypertension.**

might be the time of the study, the age group of the population studied, the diagnosis criteria for hypertension, and the study setting.

From subgroup analysis by region, the highest prevalence of hypertension (25%) was observed in the capital city of Ethiopia, Addis Ababa. This is similar to subgroup analysis by the residence which is the prevalence of hypertension (23%) was higher in urban inhabitants. The possible justification might be, urbanization may be linked to low physical activity, consumption of unhealthy diet and stress which may again leads to the high burden of non-communicable diseases [73–76].

This review also identifies the determinant factors of hypertension. In random effects model pooled estimate, sex, age, body mass index, khat chewing, alcohol consumption, family history of hypertension, and family history of diabetes mellitus were significantly associated with hypertension.

From the random effects model estimate, the pooled odds of developing hypertension among males were 29% higher than females. This finding was similar with the studies conducted in Nepal, Varanasi India, Burkina Faso, Debrecen city of Hungary, and a meta-analysis study from Vietnam [62, 69, 77–80], whereas it is not similar to a study conducted at Uganda [81]. The possible reason might be males were more vulnerable to behavioral risk factors for hypertension.

The pooled effect of age greater than 35 years was 3.6 times higher than age less than 35 years to develop hypertension, which is similar to the community-based studies conducted in Uganda, Nepal, Benin, Varanasi city India, and another city of New Delhi, India [79–83]. As well, from meta-regression analysis showed that mean age and hypertension had a direct linear relationship. Age is one of the non-modifiable risk factors for hypertension. As a result, this is the fact that cardiovascular system is strongly affected by ageing; besides, ageing causes the structural and functional changes in the blood vessels that may lead to cardiovascular morbidity and mortality [84].

This review also evidenced that individuals being overweight and/or obese were venerable to hypertension. The likelihood of developing hypertension among overweight and/or obese individuals were three times higher than normal in their body mass index. This finding is similar to the previous studies conducted in different countries [63, 77–79, 81]. Besides, a study conducted in Japan evidenced that as 1 kg/m$^2$ increase in body mass index increases the odds of developing hypertension by 23% among males and 35% among females [85]. This study strengths the fact that high body mass index increases blood cholesterol level which leads to hypertension [86]. Furthermore, this review evidenced that khat (Catha edulis) chewers were 42% more likely to develop hypertension than their counterparts which was similar to the studies conducted in Ethiopia, Yemen, and a meta-analysis study from Ethiopia [87–90]. Khat contains chemicals cathinone, cathine, and amphetamine. Cathinone is structurally related to amphetamine which increases levels of dopamine in the brain by acting on the catecholaminergic synapses [91] and increase blood pressure and heart rate [92, 93].

The pooled estimates of alcohol drinking and hypertension were statistically significant in random effects model estimate with moderate heterogeneity between studies. The odds of developing hypertension among drinkers were higher than by half as compared to non-drinkers. This finding was similar to the studies done in North American and France [94–96]. Another study evidenced that consuming three or more drinks of alcohol per day which approximately doubles the risk of developing hypertension [97]. Alcohol consumption affects the central nervous system which enhances cardiac output and has an effect on peripheral vascular effects [98].

Furthermore, family history of hypertension was a potential determinant factor for hypertension. Individuals who had a family history of hypertension have almost five times more chance to develop hypertension than individuals who had no family history of hypertension. This finding was similar to the previous studies conducted in China, Sri Lanka, and Mexico [99–101]. In addition, individuals who had a family history of diabetes mellitus were 3.7 times more likely to develop hypertension as compared to their counterparts. These factors are non-modifiable risk factors for hypertension. The possible association of family history of hypertension and diabetes mellitus with hypertension might be close blood relatives might have the same genes which may predispose to hypertension. Besides, close blood relatives might have experience of common behavioral practices that may predispose to hypertension.

This study follows some strengths and limitations. Our review adds considerable knowledge of the updated prevalence of hypertension in Ethiopia. All included studies use the same definition to declare hypertension. Subgroup analysis was performed to minimize statistical heterogeneity. Multiple factors were also included to identify the significant factors for hypertension. However, substantial statistically significant heterogeneity was observed across studies which undermine the pooled estimate of hypertension suggests that chance could be responsible for between-study variability. Sub-group analysis could not identify the source of heterogeneity. Though, meta-regression analysis suggested that mean age and region explain some source of heterogeneity.

## Conclusions and recommendations

In conclusion, hypertension is becoming a major public health problem in Ethiopia. Nearly two out of ten individuals who are older than 18 years in Ethiopia are living with hypertension. The highest prevalence of hypertension was observed in Addis Ababa and the lowest was in Tigray region. Sex, age, overweight and/or obesity, chat chewing, alcohol consumption, family history of hypertension and family history of diabetes mellitus were statistically significant factors for hypertension. Based on the finding of this review, we recommend that health planners,

policymakers, and the community itself should give prior attention to behavioral risk factors such as chat chewing, alcohol drinking and sedentary lifestyle.

## Supporting information

**S1 Table. Studies search strategies and entry terms from different electronic databases on the prevalence and determinants of hypertension.**
(DOCX)

**S1 Fig. Sensitivity analysis plot for the pooled prevalence of hypertension.**
(TIF)

**S2 Fig. Assessment of sensitivity analysis plot for factor sex.**
(TIF)

**S3 Fig. Assessment of sensitivity analysis plot for the factor age.**
(TIF)

**S4 Fig. Assessment of sensitivity analysis plot for factor among obese and/or overweight.**
(TIF)

**S5 Fig. Assessment of sensitivity analysis plot for factor Khat Chewing.**
(TIF)

**S6 Fig. Assessment of sensitivity analysis plot for factor alcohol consumption.**
(TIF)

**S7 Fig. Assessment of sensitivity analysis plot for factor family history of hypertension.**
(TIF)

**S8 Fig. Assessment of sensitivity analysis plot for factor alcohol consumption.**
(TIF)

**S1 Checklist.**
(DOC)

**S1 File.**
(XLSX)

## Author Contributions

**Conceptualization:** Sofonyas Abebaw Tiruneh, Yeaynmarnesh Asmare Bukayaw, Seblewongel Tigabu Yigizaw, Dessie Abebaw Angaw.

**Data curation:** Sofonyas Abebaw Tiruneh, Yeaynmarnesh Asmare Bukayaw, Seblewongel Tigabu Yigizaw.

**Formal analysis:** Sofonyas Abebaw Tiruneh, Yeaynmarnesh Asmare Bukayaw, Dessie Abebaw Angaw.

**Investigation:** Sofonyas Abebaw Tiruneh, Yeaynmarnesh Asmare Bukayaw, Seblewongel Tigabu Yigizaw, Dessie Abebaw Angaw.

**Methodology:** Sofonyas Abebaw Tiruneh, Yeaynmarnesh Asmare Bukayaw, Seblewongel Tigabu Yigizaw, Dessie Abebaw Angaw.

**Project administration:** Dessie Abebaw Angaw.

**Resources:** Sofonyas Abebaw Tiruneh, Yeaynmarnesh Asmare Bukayaw, Seblewongel Tigabu Yigizaw.

**Software:** Sofonyas Abebaw Tiruneh, Dessie Abebaw Angaw.

**Supervision:** Sofonyas Abebaw Tiruneh, Dessie Abebaw Angaw.

**Validation:** Sofonyas Abebaw Tiruneh, Yeaynmarnesh Asmare Bukayaw, Seblewongel Tigabu Yigizaw, Dessie Abebaw Angaw.

**Visualization:** Sofonyas Abebaw Tiruneh, Yeaynmarnesh Asmare Bukayaw, Seblewongel Tigabu Yigizaw, Dessie Abebaw Angaw.

**Writing – original draft:** Sofonyas Abebaw Tiruneh.

**Writing – review & editing:** Sofonyas Abebaw Tiruneh, Yeaynmarnesh Asmare Bukayaw, Seblewongel Tigabu Yigizaw, Dessie Abebaw Angaw.

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
