## [Decision Letter · Decision Letter 0]

11 Jun 2020

PONE-D-19-34928

Prevalence and determinants of hypertension in Ethiopia. Systematic review and meta-analysis

PLOS ONE

Dear Dr. Tiruneh,

Thank you for submitting your manuscript to PLOS ONE. After careful consideration, we feel that it has merit but does not fully meet PLOS ONE’s publication criteria as it currently stands. Therefore, we invite you to submit a revised version of the manuscript that addresses the points raised during the review process.

Multiple reviewers have made comments on this manuscript, and are interested in the theme. However, serious concerns have arisen, and each of these must be handled individually. Please indicate how each of these points have been addressed to improve the quality of the paper. 

We look forward to receiving your revised manuscript.

Kind regards,

Jay Widmer

Academic Editor

PLOS ONE

Journal Requirements:

2. Please upload a copy of Figure 1, to which you refer in your text on page 6. If the figure is no longer to be included as part of the submission please remove all reference to it within the text.

4. Please correct your reference to "p=0.000" to "p<0.001" or as similarly appropriate, as p values cannot equal zero.

Reviewers' comments:

Reviewer's Responses to Questions

**Comments to the Author**

1. Is the manuscript technically sound, and do the data support the conclusions?

Reviewer #1: Partly

Reviewer #2: Yes

2. Has the statistical analysis been performed appropriately and rigorously? 

Reviewer #1: Yes

Reviewer #2: Yes

3. Have the authors made all data underlying the findings in their manuscript fully available?

Reviewer #1: Yes

Reviewer #2: Yes

4. Is the manuscript presented in an intelligible fashion and written in standard English?

Reviewer #1: No

Reviewer #2: No

5. Review Comments to the Author

Reviewer #1: Language:

Please have the paper read and corrected by native English. Currently, language needs to be reviewed

Introduction:

Hypertension is defined and diagnosed if the systolic blood pressure readings ≥ 140 mmHg and/or the diastolic blood pressure readings ≥ 90 mmHg (1,2)

• This definition with numbers is controversial (See AHA/ACC guidelines) and does not add anything in the introduction. Authors should either adopt a conceptual definition or cancel it. The introduction could simply start by acknowledging the magnitude of the problem

Methods

• Definition of hypertension with numbers comes back and is inappropriate for such a systematic review. Authors did not defined hypertension but instead relied on definition given across studies and the variation of these defintions should be presented in tables because it could affect the results/prevalence

• Inclusion criteria. Given the differences in methodology and the large time interval between the first and the latest study, the variation of the definition of hypertension across studies, the sampling method etc…it would have been more rigorous to add a criteria like a minimum sample size to reduce study bias. Should authors have considered sample size across studies, arbitrarily I would think a minimum sample size of 1000 would be advisable. Would the results be different? Currently 308 to 9788 with HT prevalence ranging from 9 to 41%. Heterogeneity handling alone cannot control all these bias.

Results section

This systematic review and meta-analysis include published articles on the prevalence and determinates of hypertension in Ethiopia. We used PubMed, Hinari, Google Scholar, and grey literature search to find potential studies for this systematic review and meta-analysis.

• Repetition of the aim and method, not necessary

Figures

Figure 1 not provided

Choose between 5 and 7

Don’t see the added value of 10 and 11

Titles: a figure beind a stand alone, titles should be comprehensive enough. Indicating year is not really informative here. Eg, figure 1 could read : Prevalence of hypertension among people with hypertension in Ethiopia. Blue boxes represent the effect estimates (prevalence) and the horizontal bars about are for the 95% confidence intervals (CIs). The size of the boxes is proportional to the inverse variance. The diamond is for the pooled effect estimate and 95% CI, and the plain red vertical line centered on the diamond has been added to assist visual interpretation. Effect estimates are provided as proportions.

Discussion

• This pooled prevalence is low when compared to continental (Ataklte F, et al. Burden of undiagnosed hypertension in sub-saharan Africa: a systematic review and meta-analysis. Hypertension 2015; 65: 291–98.) pooled prevalence and shall be discussed.

Limitations are several:

• Studying just the prevalence, not awareness and treatment to define a response strategy to hypertension burden would be limited, one really needs to know all these components to define a response

• Non comprehensive search strategy (language bias, Only English was consider) might not have traced all studies.

• Substantial heterogeneity across the studies, this has surely affected the pooled prevalence

• Differences in individual studies’ methodology and population structures

• Methods for assessing hypertension prevalence across studies. How could any of the study use a standardize method like the WHO-STEP???

• Information on the use of antihypertensive medications is crucial because it could have affected the 140/90.

• Did authors search for unpublished data to reduce publication bias?.

Referencing

• References selection is too comprehensive but not always pertinent. Authors should read references and select the best ones, not exceeding 50 references for such a paper

• A key paper in the field is not cited: Ataklte F, Erqou S, Kaptoge S, Taye B, Echou o-Tcheugui JB, Kengne AP. Burden of undiagnosed hypertension in sub-saharan Africa: a systematic review and meta-analysis. Hypertension 2015;

65: 291–98.

Reviewer #2: General comment

This systematic review and meta-analysis is interesting in view of a rising burden of hypertension in LMICs and increasing number of publications. The analysis was handled very well and results presented appropriately. However, the written part especially in the introduction and discussions is rather sloppy. A lot of this is due to language difficulties which I’m sure can be addressed by hiring a native English speaker to edit.

I have specific comments for each section as follows:

Introduction

Apart from the common grammatical errors; the presentation of the literature on the burden needs to be re-organized in paragraphs into four main themes.

1-The Big picture (Burden of hypertension)

2- The specific problem (Literature about hypertension systematic reviews in Ethiopia and what the burden is)

3- The GAP (need to show the missing information in previous systematic reviews) and how this one fills the gap.

4- How this review fills the gap. One obvious reason is updating the previous reviews (I found 2 of them all published in 2015) and also state what you do to handle the review differently.

In the current form the is a mix of literature and sometimes contradicting information from two sources is posted but not explained example

For example the articles cited below speak about the same thing but are not brought together in a coherent way.

“Approximately 970 million people worldwide have high blood pressure and 1.56 billion adults will be living with hypertension by 2025 (3)”

“Globally around 1.13 billion people live with hypertension, which is two-thirds of them from low and middle-income countries (5).”

The statement that no systematic reviews were available to the knowledge of the authors is not true. I came across two systematic reviews on hypertension in Ethiopia and no mention of these were made apart from one that appeared in the discussion. The authors need to be honest about the scientific premise and show what gaps were in the previous review to warrant their review. See below the two reviews I came across

1-Kibret, K. T., & Mesfin, Y. M. (2015). Prevalence of hypertension in Ethiopia: a systematic meta-analysis. Public Health Reviews, 36(1), 14

2- Molla, M. (2015). Systematic reviews of prevalence and associated factors of hypertension in Ethiopia: finding the evidence. Sci J Public Health, 3(4), 514-9.

Methods

Just checking on the sensitivity of the search criteria: a synonym for high blood pressure “raised blood pressure” should have been indicated in the search terms. It is possible some articles could have been missed if they only used this terminology.

Inclusion criteria

Why did you choose 2000 as the earliest year to limit your search? Please give the rationale for selecting articles from this year.

Quality assessment

The statement below is confusing. Please clarify

“Based on the score of the quality assessment tool the highest score from nine questions declared low risk of bias” this is not clear. This would mean the highest score had the minimum risk yet the proceeding statement says the opposite

Results

You are silent about grey literature in your reporting yet you did mention an attempt to search for this in the methods section. Pease clarify if any grey literature was found and how it was handled.

Discussion

In the discussion section, a good attempt is made to interpret the findings but the language makes it rather sloppy. Need to hire the services of a native English speaker for language edits.

The comment in the limitation section about social desirability of the study is erroneous. The phrase social desirability bias is not appropriately used. I do not think cross sectional studies lead to social desirability. This could as well happen in longitudinal studies.

“Social desirability bias refers to the tendency of research subjects to give socially desirable responses instead of choosing responses that are reflective of their true feelings. The bias in responses due to this personality trait becomes a major issue when the scope of the study involves socially sensitive issues such as politics, religion, and environment, or personal issues such as drug use, cheating, and smoking. This is usually resolved by use of a well‐trained interviewer or collection of data through methods that do not require presence involvement of an interviewer can help avoid this bias to some extent. Properly identified options to questions vulnerable to social desirability effect is another means of tackling this issue”

6. PLOS authors have the option to publish the peer review history of their article (what does this mean?). If published, this will include your full peer review and any attached files.

Reviewer #1: No

Reviewer #2: No

---

## [Author Response · Author response to Decision Letter 0]

23 Jul 2020

Noted, thank you! We had tried to make it fine according to PLOS One requirements. 

2. Please upload a copy of Figure 1, to which you refer in your text on page 6. If the figure is no longer to be included aspart of the submission please remove all reference to it within the text.

Noted it was updated!

Yes, updated accordingly. 

4. Please correct your reference to "p=0.000" to "p<0.001" or as similarly appropriate, as p values cannot equal zero.

Noted, Thank you! It was updated according to the comment. 

Reviewers' comments:

Reviewer #1: Language:

Please have the paper read and corrected by native English. Currently, language needs to be reviewed

Thank you! We had tried to correct the English language intensively by reading again and again. 

Introduction:

Hypertension is defined and diagnosed if the systolic blood pressure readings ≥ 140 mmHg and/or the diastolic blood pressure readings ≥ 90 mmHg (1,2)

• This definition with numbers is controversial (See AHA/ACC guidelines) and does not add anything in the introduction. Authors should either adopt a conceptual definition or cancel it. The introduction could simply start by acknowledging the magnitude of the problem

Noted, Thank you! Accepted accordingly. 

Methods

• Definition of hypertension with numbers comes back and is inappropriate for such a systematic review. Authors did not defined hypertension but instead relied on definition given across studies and the variation of these definitions should be presented in tables because it could affect the results/prevalence

Yes! Accepted accordingly.

• Inclusion criteria. Given the differences in methodology and the large time interval between the first and the latest study, the variation of the definition of hypertension across studies, the sampling method etc…it would have been more rigorous to add a criteria like a minimum sample size to reduce study bias. Should authors have considered sample size across studies, arbitrarily I would think a minimum sample size of 1000 would be advisable. Would the results be different? Currently 308 to 9788 with HT prevalence ranging from 9 to 41%. Heterogeneity handling alone cannot control all these bias.

Thank you for the comment. We had tried to handle the heterogeneity even with sample size category by a cut point greater than 500, 1000; But the heterogeneity still their; finally, we state as limitation of the study. Besides, to include studies sample size less than 1000 might be mandatory since the maximum sample size for prevalence studies is 384 at a proportion value 50%. 

Results section

This systematic review and meta-analysis include published articles on the prevalence and determinates of hypertension in Ethiopia. We used PubMed, Hinari, Google Scholar, and grey literature search to find potential studies for this systematic review and meta-analysis.

• Repetition of the aim and method, not necessary

Thank you! corrected accordingly.

Figures

Figure 1 not provided

Thank you, corrected accordingly. 

Choose between 5 and 7

Don’t see the added value of 10 and 11

Titles: a figure behind a standalone, titles should be comprehensive enough. Indicating year is not really informative here. Eg, figure 1 could read: Prevalence of hypertension among people with hypertension in Ethiopia. Blue boxes represent the effect estimates (prevalence) and the horizontal bars about are for the 95% confidence intervals (CIs). The size of the boxes is proportional to the inverse variance. The diamond is for the pooled effect estimate and 95% CI, and the plain red vertical line centered on the diamond has been added to assist visual interpretation. Effect estimates are provided as proportions.

Thank you! Corrected accordingly.

Discussion

o This pooled prevalence is low when compared to continental (Ataklte F, et al. Burden of undiagnosed hypertension in sub-saharan Africa: a systematic review and meta-analysis. Hypertension 2015; 65: 291–98.) pooled prevalence and shall be discussed.

Thank you! Corrected and cited.

Limitations are several:

o Studying just the prevalence, not awareness and treatment to define a response strategy to hypertension burden would be limited, one really needs to know all these components to define a response

o Non comprehensive search strategy (language bias, Only English was consider) might not have traced all studies.

o Substantial heterogeneity across the studies, this has surely affected the pooled prevalence

o Differences in individual studies’ methodology and population structures

o Methods for assessing hypertension prevalence across studies. How could any of the study use a standardize method like the WHO-STEP???

o Information on the use of antihypertensive medications is crucial because it could have affected the 140/90.

o Did authors search for unpublished data to reduce publication bias?

Noted, accepted thank you! We, act accordingly the comment.

Referencing

• References selection is too comprehensive but not always pertinent. Authors should read references and select the best ones, not exceeding 50 references for such a paper

• A key paper in the field is not cited: Ataklte F, Erqou S, Kaptoge S, Taye B, Echou o-Tcheugui JB, Kengne AP. Burden of undiagnosed hypertension in sub-saharan Africa: a systematic review and meta-analysis. Hypertension 2015; 65: 291–98.

Noted!

Reviewer #2: General comment

This systematic review and meta-analysis is interesting in view of a rising burden of hypertension in LMICs and increasing number of publications. The analysis was handled very well and results presented appropriately. 

Noted, Thank you!

However, the written part especially in the introduction and discussions is rather sloppy. A lot of this is due to language difficulties which I’m sure can be addressed by hiring a native English speaker to edit.

Noted, we have read again and again to improve the quality of the English writing roblem, as well we had invited for English editors in our University. 

I have specific comments for each section as follows:

Introduction

Apart from the common grammatical errors; the presentation of the literature on the burden needs to be re-organized in paragraphs into four main themes. 

1. The Big picture (Burden of hypertension).

2. The specific problem (Literature about hypertension systematic reviews in Ethiopia and what the burden is)

3. The GAP (need to show the missing information in previous systematic reviews) and how this one fills the gap.

4. How this review fills the gap. One obvious reason is updating the previous reviews (I found 2 of them all published in 2015) and state what you do to handle the review differently.

Noted, thank you very much for this wonderful comment and it was updated accordingly. 

In the current form the is a mix of literature and sometimes contradicting information from two sources is posted but not explained example

For example the articles cited below speak about the same thing but are not brought together in a coherent way. “Approximately 970 million people worldwide have high blood pressure and 1.56 billion adults will be living with hypertension by 2025 (3)”

“Globally around 1.13 billion people live with hypertension, which is two-thirds of them from low and middle-income countries (5).”

Noted, Thank you! Corrected accordingly. 

The statement that no systematic reviews were available to the knowledge of the authors is not true. I came across two systematic reviews on hypertension in Ethiopia and no mention of these were made apart from one that appeared in the discussion. The authors need to be honest about the scientific premise and show what gaps were in the previous review to warrant their review. See below the two reviews I came across

1-Kibret, K. T., & Mesfin, Y. M. (2015). Prevalence of hypertension in Ethiopia: a systematic meta-analysis. Public Health

Reviews, 36(1), 14

2- Molla, M. (2015). Systematic reviews of prevalence and associated factors of hypertension in Ethiopia: finding the evidence. Sci J Public Health, 3(4), 514-9.

Noted, Thank you! Yes, there are two reviews by the year 2015. But these two reviews did not account factors that affect the pooled prevalence of hypertension, as well as 20 (52.60%) studies, had been published after 2016. Therefore, this review gives updated information for interventions. 

Methods

Just checking on the sensitivity of the search criteria: a synonym for high blood pressure “raised blood pressure” should have been indicated in the search terms. It is possible some articles could have been missed if they only used this terminology.

Noted, thank you! The searching strategies was performed accordingly in the MeSH terms using the key terms of hypertension. Therefore, the synonym was performed accordingly the MeSH terms by hypertension. 

Inclusion criteria

Why did you choose 2000 as the earliest year to limit your search? Please give the rationale for selecting articles from this year.

Noted, thank you! We interested updated information in the last two decades simply. As well as no relevant studies before 2009 for this review. 

Quality assessment

The statement below is confusing. Please clarify

“Based on the score of the quality assessment tool the highest score from nine questions declared low risk of bias” this is not clear. This would mean the highest score had the minimum risk yet the proceeding statement says the opposite

Noted, and updated accordingly!

Results

You are silent about grey literature in your reporting yet you did mention an attempt to search for this in the methods section. Pease clarify if any grey literature was found and how it was handled.

 Noted, Thank you! Yes, we had include one grey literature (Birhanu Tolera (58)) and it was extracted the same way as the other studies. 

Discussion

In the discussion section, a good attempt is made to interpret the findings but the language makes it rather sloppy. Need to hire the services of a native English speaker for language edits.

Noted thank you corrected accordingly. 

The comment in the limitation section about social desirability of the study is erroneous. The phrase social desirability bias is not appropriately used. I do not think cross sectional studies lead to social desirability. This could as well happen in longitudinal studies.

“Social desirability bias refers to the tendency of research subjects to give socially desirable responses instead of choosing responses that are reflective of their true feelings. The bias in responses due to this personality trait becomes a major issue when the scope of the study involves socially sensitive issues such as politics, religion, and environment, or personal issues such as drug use, cheating, and smoking. This is usually resolved by use of a well‐trained interviewer or collection of data through methods that do not require presence involvement of an interviewer can help avoid this bias to some extent. Properly identified options to questions vulnerable to social desirability effect is another means of tackling this issue”

Thank you very much for the detail elaboration. Yes, we are go through the limitation and correct according to the comment and that was account.

---

## [Decision Letter · Decision Letter 1]

13 Aug 2020

PONE-D-19-34928R1

Prevalence and determinants of hypertension in Ethiopia. Systematic review and meta-analysis

PLOS ONE

Dear Dr. Tiruneh,

Thank you for submitting your manuscript to PLOS ONE. After careful consideration, we feel that it has merit but does not fully meet PLOS ONE’s publication criteria as it currently stands. Therefore, we invite you to submit a revised version of the manuscript that addresses the points raised during the review process.

We look forward to receiving your revised manuscript.

Kind regards,

Jay Widmer

Academic Editor

PLOS ONE

Reviewers' comments:

Reviewer's Responses to Questions

**Comments to the Author**

1. If the authors have adequately addressed your comments raised in a previous round of review and you feel that this manuscript is now acceptable for publication, you may indicate that here to bypass the “Comments to the Author” section, enter your conflict of interest statement in the “Confidential to Editor” section, and submit your "Accept" recommendation.

Reviewer #2: All comments have been addressed

2. Is the manuscript technically sound, and do the data support the conclusions?

Reviewer #2: Partly

3. Has the statistical analysis been performed appropriately and rigorously? 

Reviewer #2: Yes

4. Have the authors made all data underlying the findings in their manuscript fully available?

Reviewer #2: Yes

5. Is the manuscript presented in an intelligible fashion and written in standard English?

Reviewer #2: No

6. Review Comments to the Author

Reviewer #2: The language edits are not satisfactory. I doubt if the authors sought services of a native English speaker. I noted several grammatical errors. It appears to me more errors were introduced during the revisions. I have cited some of them here but this is not an exhaustive list. There will be need for a thorough check. I strongly recommend hiring a language editor.

Introduction

Lines 58-60- According to the World Health Organization (WHO) report in 2019, globally more than 1.13 billion people live with hypertension, of this two-thirds of them from Low and Middle-Income Countries (LMICs)

Lines 63-64. According to a systematic study, the overall pooled prevalence of hypertension in Africa raised from 19.7% in 1990, 27.4% in 2000 and 30.8% in 2010 [5].

Line 65: people live with hypertension and it will be projected 3 out of 4 people by the end of 2025

Lines 76-77: increment of non-communicable diseases will lead to greater dependency and rise the costs of health care for patients and their families unless public health interventions.

Line 82: Besides, more than half of the studies published after the previous study conducted.

Discussion

Line 305: Non-communicable disease was a double burden of public health problem in developing countries

Line 330: This finding was consistence with a study conducted in Nepal

7. PLOS authors have the option to publish the peer review history of their article (what does this mean?). If published, this will include your full peer review and any attached files.

Reviewer #2: No

---

## [Author Response · Author response to Decision Letter 1]

15 Sep 2020

Response to Reviewers’

Prevalence and determinants of hypertension in Ethiopia: Systematic review and meta-analysis

Sofonyas Abebaw Tiruneh1, Yeaynmarnesh Asmare Bukayaw2, Seblewongel Tigabu Yigizaw2, Dessie Abebaw Angaw2 

The authors, extend great thanks for the editors and reviewers as the stand of this review. The comments raised by the reviewers are vital and defiantly it will improve the quality of the manuscript. 

Stay Safe!!!

The Authors.

Reviewer #2: The language edits are not satisfactory. I doubt if the authors sought services of a native English speaker. I noted several grammatical errors. It appears to me more errors were introduced during the revisions. I have cited some of them here but this is not an exhaustive list. There will be need for a thorough check. I strongly recommend hiring a language editor.

Introduction

Lines 58-60- According to the World Health Organization (WHO) report in 2019, globally more than 1.13 billion people live with hypertension, of this two-thirds of them from Low and Middle-Income Countries (LMICs)

Lines 63-64. According to a systematic study, the overall pooled prevalence of hypertension in Africa raised from 19.7% in 1990, 27.4% in 2000 and 30.8% in 2010 [5].

Line 65: people live with hypertension and it will be projected 3 out of 4 people by the end of 2025

Lines 76-77: increment of non-communicable diseases will lead to greater dependency and rise the costs of health care for patients and their families unless public health interventions.

Line 82: Besides, more than half of the studies published after the previous study conducted.

Discussion

Line 305: Non-communicable disease was a double burden of public health problem in developing countries

Line 330: This finding was consistence with a study conducted in Nepal

We noted all the concerns and we have tried to our maximum effort to improve the quality of English writing.

We also invited the English exert river in our institution.

---

## [Editor Report · Decision Letter 2]

23 Sep 2020

PONE-D-19-34928R2

Prevalence and determinants of hypertension in Ethiopia. Systematic review and meta-analysis

PLOS ONE

Dear Dr. Tiruneh,

Thank you for submitting your manuscript to PLOS ONE. After careful consideration, we feel that it has merit but does not fully meet PLOS ONE’s publication criteria as it currently stands. Therefore, we invite you to submit a revised version of the manuscript that addresses the points raised during the review process.

The reviewers feel that the editorial changes were not satisfactory to merit publication. Particularly, the authors should pay particular attention to each and every sentence to be sure it has been properly proofed and vetted by a native English speaker for correct grammar and syntax. Without these changes we will be unable to publish this work that has some merit according to the reviewers and editorial staff. Please make these changes at your earliest convenience and resubmit. Thank you.

We look forward to receiving your revised manuscript.

Kind regards,

Jay Widmer

Academic Editor

PLOS ONE

---

## [Author Response · Author response to Decision Letter 2]

20 Oct 2020

Editors request 

The reviewers feel that the editorial changes were not satisfactory to merit publication. Particularly, the authors should pay particular attention to each and every sentence to be sure it has been properly proofed and vetted by a native English speaker for correct grammar and syntax. Without these changes we will be unable to publish this work that has some merit according to the reviewers and editorial staff. Please make these changes at your earliest convenience and resubmit. 

Authors response

The authors, extend great thanks for the editors and reviewers as the stand of this review. The comments raised by the editors are valuable and defiantly it will improve the quality of the manuscript. We have addressed the English and grammar problems as per the editor request.

Since all authors are from low-income country, we cannot able to proofread the manuscript by fluent in English by payment especially native by English. But we have tried to improve the grammar and punctuation problems through read again and again. Besides, we had invited the manuscript for senior English editors at the University of Gondar and Debre Tabor University, Ethiopia. Those two English experts Lectures review our manuscript by free cost. 

We hope that the English write up problem improves and will meet the minimum publication criteria of PLOS ONE journal. We look forwarding your excellent comments as usual. 

The authors,

---

## [Decision Letter · Decision Letter 3]

10 Nov 2020

PONE-D-19-34928R3

Prevalence and determinants of hypertension in Ethiopia. Systematic review and meta-analysis

PLOS ONE

Dear Dr. Tiruneh,

Thank you for submitting your manuscript to PLOS ONE. After careful consideration, we feel that it has merit but does not fully meet PLOS ONE’s publication criteria as it currently stands. Therefore, we invite you to submit a revised version of the manuscript that addresses the points raised during the review process.

The edits are not acceptable as written, and careful attention to correcting the text to ensure readability in the English language is required for further consideration for publication. 

We look forward to receiving your revised manuscript.

Kind regards,

R. Jay Widmer

Academic Editor

PLOS ONE

Additional Editor Comments (if provided):

The scientific merit has certainly been met, however the language simply is illegible by current English standards. This will be the authors final chance to amend the paper so that all grammatical mistakes are corrected and the paper flows in a smooth and readable fashion. For example the second sentence of the abstract reads, "A comprehensive electronic databases". There is a lack of subject/verb agreement, and this phrase is uninterpretable by the reader. Please make this final effort to correct the paper, and we look forward to a thoroughly reviewed next draft. Thank you for your attention to these details.

Reviewers' comments:

Reviewer's Responses to Questions

2. Is the manuscript technically sound, and do the data support the conclusions?

Reviewer #1: Yes

3. Has the statistical analysis been performed appropriately and rigorously? 

Reviewer #1: I Don't Know

4. Have the authors made all data underlying the findings in their manuscript fully available?

Reviewer #1: Yes

6. Review Comments to the Author

Reviewer #1: I very much appreciate the efforts to address all my comments. However,

1. You probably did not get my comment on the introduction, I would suggest you simply cancel the first sentence of the introduction, it is controversial and adds nothing.

2. The language edits are really far below what is acceptable. My suggestion would be that you have the paper read by a scientific writer.

7. PLOS authors have the option to publish the peer review history of their article (what does this mean?). If published, this will include your full peer review and any attached files.

Reviewer #1: No

---

## [Author Response · Author response to Decision Letter 3]

12 Dec 2020

Response Letter to editors and reviewers 

Prevalence of hypertension and its determinants in Ethiopia: A systematic review and meta-analysis

Sofonyas Abebaw Tiruneh1*, Yeaynmarnesh Asmare Bukayaw2, Seblewongel Tigabu Yigizaw2, Dessie Abebaw Angaw2

We authors appreciate to the editors and reviewers for this manuscript as the position fourth review. The comments raised by the editors and reviewers are vital and defiantly it will improve the quality of the manuscript. We have addressed all the issues raised by the editors and reviewer's and believed that the revised version of the manuscript is satisfactory and will meet the minimum journal publication requirements. We have updated the manuscript accordingly. 

Stay Safe!!!

The Authors.

Additional Editor Comments (if provided):

The scientific merit has certainly been met, however, the language simply is illegible by current English standards. This will be the authors final chance to amend the paper so that all grammatical mistakes are corrected and the paper flows in a smooth and readable fashion. For example the second sentence of the abstract reads, "A comprehensive electronic databases". There is a lack of subject/verb agreement, and this phrase is uninterpretable by the reader. Please make this final effort to correct the paper, and we look forward to a thoroughly reviewed next draft. Thank you

for your attention to these details.

Noted: Thank you! We have made our final attempt as our maximum effort.

 Review Comments to the Author

Reviewer #1: I very much appreciate the efforts to address all my comments. However,

1. You probably did not get my comment on the introduction, I would suggest you simply cancel the first sentence of the introduction, it is controversial and adds nothing.

2. The language edits are really far below what is acceptable. My suggestion would be that you have the paper read by a scientific writer.

Noted thank you, We appreciate your comment and We had corrected in the recent version of the manuscript

---

## [Editor Report · Decision Letter 4]

15 Dec 2020

Prevalence of hypertension and its determinants in Ethiopia: A systematic review and meta-analysis

PONE-D-19-34928R4

Dear Dr. Tiruneh,

We’re pleased to inform you that your manuscript has been judged scientifically suitable for publication and will be formally accepted for publication once it meets all outstanding technical requirements.

Kind regards,

R. Jay Widmer

Academic Editor

PLOS ONE

Additional Editor Comments (optional):

The authors are to be congratulated on this work and for such a well written manuscript.

---

## [Editor Report · Acceptance letter]

18 Dec 2020

PONE-D-19-34928R4 

Prevalence of hypertension and its determinants in Ethiopia: A systematic review and meta-analysis 

Dear Dr. Tiruneh:

I'm pleased to inform you that your manuscript has been deemed suitable for publication in PLOS ONE. Congratulations! Your manuscript is now with our production department. 

Kind regards, 

on behalf of

Dr. R. Jay Widmer 

Academic Editor

PLOS ONE